# Enhanced Resistance of *atbzip62* against *Pseudomonas syringae* pv. *tomato* Suggests Negative Regulation of Plant Basal Defense and Systemic Acquired Resistance by *AtbZIP62* Transcription Factor

**DOI:** 10.3390/ijms222111541

**Published:** 2021-10-26

**Authors:** Rizwana Begum Syed Nabi, Nkulu Kabange Rolly, Rupesh Tayade, Murtaza Khan, Muhammad Shahid, Byung-Wook Yun

**Affiliations:** 1Laboratory of Plant Functional Genomics, School of Applied Biosciences, Kyungpook National University, Daegu 41566, Korea; ruhii.syed@gmail.com (R.B.S.N.); rolly.kabange@gmail.com (N.K.R.); murtazakhan.bio@gmail.com (M.K.); shahidariswat@gmail.com (M.S.); 2Department of Southern Area Crop Science, National Institute of Crop Science, RDA, Miryang 50424, Korea; 3National Laboratory of Seed Testing, National Seed Service, SENASEM, Ministry of Agriculture, Kinshasa 904KIN1, Democratic Republic of the Congo; 4Laboratory of Plant Breeding, School of Applied Biosciences, Kyungpook National University, Daegu 41566, Korea; rupesh.tayed@gmail.com; 5Agriculture Research Institute Mingora, Swat 19130, Khyber Pakhtunkhwa, Pakistan

**Keywords:** disease resistance, *Pseudomonas syringae* pv. *tomato*, *AtbZIP62* TF, systemic acquired resistance, *Arabidopsis*

## Abstract

The intrinsic defense mechanisms of plants toward pathogenic bacteria have been widely investigated for years and are still at the center of interest in plant biosciences research. This study investigated the role of the *AtbZIP62* gene encoding a transcription factor (TF) in the basal defense and systemic acquired resistance in Arabidopsis using the reverse genetics approach. To achieve that, the *atbzip62* mutant line (lacking the *AtbZIP62* gene) was challenged with *Pseudomonas syringae* pv. *tomato* (*Pst DC3000*) inoculated by infiltration into Arabidopsis leaves at the rosette stage. The results indicated that *atbzip62* plants showed an enhanced resistance phenotype toward *Pst DC3000 vir* over time compared to Col-0 and the susceptible disease controls, *atgsnor1-3* and *atsid2.* In addition, the transcript accumulation of pathogenesis-related genes, *AtPR1* and *AtPR2*, increased significantly in *atbzip62* over time (0–72 h post-inoculation, hpi) compared to that of *atgsnor1-3* and *atsid2* (susceptible lines), with *AtPR1* prevailing over *AtPR2*. When coupled with the recorded pathogen growth (expressed as a colony-forming unit, CFU mL^−1^), the induction of *PR* genes, associated with the salicylic acid (SA) defense signaling, in part explained the observed enhanced resistance of *atbzip62* mutant plants in response to *Pst DC3000 vir*. Furthermore, when *Pst DC3000 avrB* was inoculated, the expression of *AtPR1* was upregulated in the systemic leaves of Col-0, while that of *AtPR2* remained at a basal level in Col-0. Moreover, the expression of *AtAZI* (a systemic acquired resistance -related) gene was significantly upregulated at all time points (0–24 h post-inoculation, hpi) in *atbzip62* compared to Col-0 and *atgsnor1-3* and *atsid2*. Under the same conditions, *AtG3DPH* exhibited a high transcript accumulation level 48 hpi in the *atbzip62* background. Therefore, all data put together suggest that *AtPR1* and *AtPR2* coupled with *AtAZI* and *AtG3DPH*, with *AtAZI* prevailing over *AtG3DPH*, would contribute to the recorded enhanced resistance phenotype of the *atbzip62* mutant line against *Pst DC3000*. Thus, the *AtbZIP62* TF is proposed as a negative regulator of basal defense and systemic acquired resistance in plants under *Pst DC3000* infection.

## 1. Introduction

Plants are sessile organisms, and because of this nature, they are always subjected to various stresses caused by environmental factors (abiotic stress) or living organisms (biotic stress), including bacterial pathogens’ attacks, among others, which can cause plant growth failure [1].

Unlike mammals or vertebrates, plants lack mobile defender cells and a somatic adaptive immune system to counterattack pathogens infection. Rather, plants rely on the innate immune system to sense and transduce the signals and respond to pathogen infections. The dynamics of plants’ defense against pathogens include the basal defense, *R* gene (or effector)-triggered immunity (ETI) [1,2], and the pathogen- or microbe-associated molecular patterns (PAMPs/MAMPs-triggered immunity, PTI). One of the earliest responses to attempted pathogen attack is the generation of oxidative burst, which may trigger a hypersensitive response (HR) and induce programmed cell death (PCD) at the infection site [3]. A HR is commonly associated with a resistance response and is regulated by direct or indirect interactions between avirulence proteins from pathogen and resistance proteins from the plant, and it may be the result of multiple signaling pathways. In addition to the basal defense, plants employ another defense strategy by involving pattern recognition receptors (PRR, transmembrane proteins) that recognize PAMPs/MAMPs, and induce a downstream signaling cascade, which activates the basal resistance mechanism [4]. It is said that PTI can be suppressed by a category of pathogen-encoded effector proteins widely recognized as avirulence (*avr*) factors [5,6], which are, in turn, recognized by host-encoded resistance (R) proteins that confer a durable and robust resistance called *R* gene- or effector-triggered immunity (ETI) [7,8,9].

As with other biotic stresses, the main impact of pathogen attacks is the generation of reactive oxygen species (ROS) and nitrogen species (RNS). When they accumulate at low concentrations, ROS and RNS serve as signaling molecules during abiotic or biotic stress. However, the over accumulation of ROS or RNS causes oxidative stress, nitro-oxidative stress, which may result in oxidative damage and cell membrane degradation, and the induction of programmed cell death (PCD) that may culminate in cell death.

As part of the various defense signaling cascades, plants respond to pathogens using the systemic acquired resistance (SAR), a salicylic acid (SA)-dependent response, also known as a long-distance signaling mechanism that provides broad-spectrum and long-lasting resistance to secondary infections across the plant [10]. The interplay between the innate immune system and antioxidant (enzymatic and nonenzymatic) systems, coupled with a transcriptional reprogramming within the cell, determine the degree of resistance and alleviate the detrimental effects of ROS and RNS and their derivative compounds [2].

Transcription factors (TFs) are regulators of gene expression in biological systems. TFs interact with other proteins or DNA in the promoter region of their target genes to regulate their expression under specific conditions [11]. TFs are said to operate either alone or in complex with other molecules to induce or suppress the recruitment of the basal transcriptional machinery to specific genes [12,13,14,15]. TFs such as basic leucine zipper (bZIP) are involved in seed germination, seed development, plant developmental processes, carbohydrate metabolism, and hormonal signaling events during abiotic and biotic stresses [16,17,18,19]. Reports have shown that, of the 75 members of the bZIP family identified in the *Arabidopsis thaliana* genome (herein referred to *Arabidopsis*) [13,20], about 50 TFs remain totally or partially functionally uncharacterized [21,22]. However, a few genes have been functionally characterized, including the *AtbZIP62* TF. The role of *AtbZIP62* in the adaptive response mechanism toward abiotic stress such as drought [12,23] and salinity stress [24] has been previously investigated, and a possible transcriptional interaction with other bZIP TF-encoding genes has been suggested.

Generally, abiotic and biotic stress response mechanisms evolve a contrasting cellular metabolism and differential physiological processes, molecular functions, signaling pathways, and biochemical reactions [20,25]. Therefore, this study investigated the role of the *AtbZIP62* TF in the defense mechanisms against the bacterium *Pseudomonas syringae* pv. *tomato* (*Pst*). To achieve that, we challenged the Arabidopsis mutant lines *atbzip62* (lacking the *AtbZIP62* gene) and Col-0 wild type (WT) with the *Pst DC3000* virulent strain over time to assess the basal defense. Additionally, we explored the possibility for the *AtbZIP62* to be associated with SAR and *R*-gene-mediated resistance toward the avirulent *Pst DC3000 avrB* strain. The Arabidopsis mutant line, *atgsnor1-3* and *atsid2*, were included in the study as disease-susceptible controls.

## 2. Results

### 2.1. Enhanced Phenotypic Resistance of atbzip62 toward Pst DC3000 vir

The *AtbZIP62* was previously reported as a positive regulator of drought stress response [12], while being suggested as a negative regulator of salt overly sensitive signaling pathway genes under salt stress [24]. In addition, *AtbZIP62* was earlier proposed to have a transcriptional interplay with some of the key hormonal signaling pathway genes under drought stress conditions [12,26]. In this study, the results of the pathogenicity test (Figure 1A) revealed that no *Pst* typical symptoms could be detected in *atbizip62-*inoculated plants 4 days post-inoculation (dpi), and slight chlorotic-like symptoms were apparent at the final count (9 dpi). By contrast, Col-0 (WT) and the disease-susceptible controls, *atgsnor1-3* (lacking the *S-*nitrosoglutathione reductase 1 (*GNSOR1*), which regulates the cellular *S-*nitrosothiols (SNO) levels) [27] and *atsid2* (SA-deficient mutant), showed chlorotic symptoms as early as 72 h after inoculation. From the recorded resistance phenotype of *atbizip62* plants, we expected to see a reduced pathogen growth in this mutant in response to *Pst DC3000 vir*. Our data in panel B of Figure 1 indicated that the bacterial growth (expressed as logarithm (log) of the colony-forming unit, CFU mL^−1^) was weak in *atbzip62* compared with that of Col-0 WT at 1 dpi while showing a similar growth pattern with that of *atsid2*. However, with the passage of time (48–72 hpi), Col-0 and *atbzip62* plants showed similar pathogen growth patterns, while *atgsnor1-3* and *atsid2* (susceptible mutant lines) exhibited a rapid proliferation of the pathogen under the same conditions.

Based on the resistance phenotype exhibited by *atbzip62* in response to *Pst DC3000 vir*, we were interested to see the transcriptional level of a set of pathogenesis-related genes in all tested genotypes (WT and mutant lines), including *AtPR1* and *AtPR2*, widely known for their important roles in the innate immune system of plants during pathogens attack. Results in panel C of Figure 1 indicate that the expression of *AtPR1* was upregulated over time in Col-0, and reached its peak at 72 hpi. When measured in *atbzip62* plants, *AtPR1* showed a similar increasing transcript accumulation pattern to that observed in Col-0, but was much lower in the mutant line. The expression of *AtPR1* in *atgsnor1-3* was significantly upregulated at 48–72 hpi, while in *atsid2*, a basal expression level was observed soon after inoculation (0–48 hpi), but the transcript accumulation of *AtPR1* increased significantly at 72 hpi. In the same way, the transcript accumulation of *AtPR2* gradually increased in Col-0 and *atbzip62* with the exposure time, with *atzip62* showing a significant upregulation soon after inoculation (24 h) compared to Col-0. Under the same conditions, the expression of *AtPR2* in the *atbzip62* mutant line was the highest among all tested genotypes over time and reached its peak 48–72 h after inoculation with *Pst DC3000 vir* (Figure 1D).

### 2.2. AtbZIP62 TF Negatively Regulates Systemic Acquired Resistance in Arabidopsis

Although plants lack an immune system comparable to animals, they have developed a stunning array of structural, chemical, and protein-based defenses designed to detect invading organisms and restrict them right before they can cause extensive damage. The ability of the plant to intercept and localize an invading pathogen is crucial for the plant to activate the appropriate defense mechanism to combat the pathogen [28]. Unlike the basal defense, SAR (also known as an SA-dependent response) is a long-distance signaling mechanism that provides a broad-spectrum and long-lasting resistance to secondary infections at the whole plant level [10,29]. Here, to examine the possibility for the *AtbZIP62* TF to be involved in the signaling network upon bacterial pathogen infection, as well as in *R* gene-mediated resistance, we inoculated plants with *Pst DC3000 avrB*. The results indicate that the transcript accumulation of *AtPR1* and *AtPR2* reached the highest level at 24–48 h after inoculation in Col-0 and *atbzip62*, while in *atgsnor1-3*, an increase was observed soon after bacterial infection 3 hpi and 24 hpi. However, *atsid2* recorded a low transcript accumulation of *AtPR1* over time (Figure 2A). From panel B of Figure 2, we could see that the pattern of *AtPR2* had a similar expression pattern with that of *AtPR1* in Col-0. Under the same conditions, *AtPR2* recorded the highest transcript accumulation level 12 h after inoculation in *atbzip62*, and gradually increased over time in a similar manner as *atgsnor1-3*, *atsid2*, and Col-0 (Figure 2B).

To further understand the observed mechanism underlying the enhanced resistance phenotype of *atbzip62* against *Pst DC3000 vir*, we analyzed the integrity of the cell membrane by measuring the electrolyte leakage in the target mutant compared to the WT and susceptible control lines. As indicated in panel C of Figure 2, the high NO-producing mutant line, *atgsnor1-3*, had the highest enhanced electrolyte leakage over time, followed by the SA-deficient mutant *atsid2*, *atbzip62*, and Col-0.

### 2.3. atbzip62 Shows Differential Response to Systemic Acquired Resistance

The systemic acquired resistance (SAR) is an SA-dependent response, also known as a long-distance signaling mechanism that provides a broad-spectrum and long-lasting resistance to secondary infections across the plant [10,30]. Therefore, we analyzed whether the *AtbZIP62* TF could have contributed to SAR in response to *Pst DC3000 avrB*. Therefore, we measured the expressions of pathogenesis-related genes (*AtPR1* and *AtPR2*), glyceraldehyde 3-phosphate dehydrogenase (*AtG3PDH*), and azelaic acid inducer (*AtAZI*) in systemic or distal leaves (unwounded leaves located away from the inoculated leaf but on the same plant). In the previous paragraphs, we observed that the transcriptional level of *AtPR1* reached its highest level at 48 h in the local leaves (treated or inoculated leaves) of Col-0 (Figure 2A). However, when measured in the systemic leaves (or distal leaves) of *atbzip62*, *AtPR1* recorded its highest expression level soon after bacterial infection (12–24 h) (Figure 3A). Furthermore, *AtPR1* transcript accumulation in the systemic leaves of *atgsnor1-3* showed an odd expression pattern but remained at a relatively low level in the systemic leaves of *atsid2* under the same conditions. In the same way, the expression of *AtPR2* was significantly upregulated soon after bacterial inoculation (0–24 hpi) in systemic leaves of *atbzip62* compared to Col-0 (Figure 3B). However, no significant difference was found in *atsid2* and *atgsnor1-3* (except at 12 h). From another perspective, we were also interested to see the expression of some of the key genes associated with SAR during pathogens attack. Our data indicate that the expression of the *AtG3DPH*, one of the key marker genes regulating SAR-response in the plants, was higher at 48 hpi in the systemic leaves of *atbzip62* compared to other genotypes (Figure 3D). Similarly, the transcript accumulation of *AtAZI*, another important SAR-related gene, was higher in *atbzip62* over time compared with those in Col-0, *atgsnor1-3*, and *atsid2* (Figure 3C).

### 2.4. In Silico Transcription Factor Binding Sites Analysis Identified Cis-Regulatory Elements for bZIP TFs in AtPR2 and AtAZI SAR-Related Gene

In previous paragraphs, we showed that the *atbzip62* mutant exhibited an enhanced resistance phenotype against *Pst DC3000 vir* over time. Thus, in addition to the transcript accumulation levels of the selected pathogenesis-related genes and SAR-marker genes, we were interested in exploring the possible interactions that AtbZIP62 may have with the target genes. The results of the transcription factor binding sites analysis detected the presence of *cis*-regulatory elements specific to two bZIP TFs, AtbZIP18 and AtbZIP69, in the promoter of the target PR and SAR-related genes. As indicated in Table 1, two binding sites for AtbZIP69 were detected in the promoter of *AtPR2*. Similarly, two binding sites for AtbZIP69 and one for AtbZIP18 were found in the promoter of *AtAZI*. The breakthrough is that AtbZIP18 and AtbZIP69 were shown to have a common binding site (with the same motif) at one of the two binding sites detected for *AtbZIP69*. However, no binding sites for AtbZIP18 or AtbZIP69 could be identified in the promoter of *AtPR1* or *AtG3PDH*.

Furthermore, panels A and B of Figure 4 proposed a possible protein–protein interaction between *AtPR1* and *AtPR2* among other PR genes. No protein–protein interaction was proposed between AtbZIP62 and both AtPR1 or AtPR2 (also known as beta-1, 3-glucanase 2 or BGL2); whereas, panels A and B of Figure 4 propose that AtPR1 would have a protein-protein interaction with AtPR2, or vice versa, among other PR proteins.

## 3. Discussion

### 3.1. Differential Phenotypic Response between atbzip62 and Col-0 Plants in Response to Pst DC3000 vir

Many pathways involved in the plant immune system are activated upon bacterial pathogens infection, including the induction of various pathogenesis-related genes and signaling cascades [31,32,33]. In the process, negative or positive plant defense regulators against pathogens are activated or suppressed. Their crosstalk determines the degree of resistance required for the plant to trigger the innate immune system [34,35,36]. Here, we inoculated the *Pst DC3000 vir* strain into *Arabidopsis atbzip62* plants lacking the *AtbZIP62* TF-encoding gene over time to assess the phenotypic response of mutant toward this bacterial pathogen, along with the Col-0 WT and two disease-susceptible mutant lines, *atgsnor1-3* and *atsid2*. Previously, in our studies, *atbzip62* plants recorded earlier susceptible phenotypic responses to abiotic stress (salinity [24] and drought stress [12]), whereas the *AtbZIP62* TF was suggested to be a positive regulator of abiotic stress response in *Arabidopsis.* Generally, genotypes that show a sensitive phenotypic response toward abiotic stress are expected to show an opposite (resistance) phenotype under biotic stress occurrence [9]. This study showed that when *atbzip62* plants were subjected to bacterial pathogen infection, this mutant exhibited a high-resistance phenotype compared to Col-0 WT, with minor *Pst* typical symptoms observed on the inoculated leaves (Figure 1A) Additionally, the bacterial CFU counts also displayed a reduced pathogen growth pattern soon after inoculation (24 h) (Figure 1B).

### 3.2. Expression Patterns of PR Genes in atbzip62 under Pst DC3000Infection Suggest Negative Regulation of Plant Basal Defense by AtbZIP62 TF

When plants are challenged with pathogenic bacteria, they activate their defense mechanisms, which include the induction of pathogenesis-related genes, such as *PR1* and *PR2*, along with various signaling cascades. Under these conditions, negative or positive regulators of plant defense against pathogens attack are suppressed or induced, and their crosstalk determines the level of resistance needed for the PTI system [1,8]. Thus, the recorded significant upregulation of *AtPR1* (24–72 hpi, Figure 1C) and *AtPR2* (24–72 hpi), with *AtPR1* prevailing over *AtPR2*, coupled with the pathogen growth pattern (Figure 1B) in *atbizp62* soon after *Pst DC3000 vir* inoculation would partly explain the observed enhanced degree of resistance of *atbzip62* mutant plants.

A previous study reported specific genes belonging to the *bZIP* TF family as being involved in regulating plant defense against biotic stress [37]. Similarly, a *bZIP* member *OsTGA2.1* was suggested to be involved in the basal defense against bacterial pathogens in rice [18]. Similarly, another study suggested that the *Os**bZIP1* TF is a positive regulator of basal defense against *Magnaporthe grisea* in rice [38].

To investigate the role of *AtbZIP62* in *R* gene-mediated resistance, we inoculated plants with *Pst DC3000 avrB*. The results showed that soon after inoculation (12–48 hpi), *AtPR1* showed a high transcript accumulation level in *atbzip62* compared with the susceptible mutant lines, *atgsnor1-3* and *atsid2*, and much less in Col-0 (Figure 3B). The expression pattern of *AtPR2* was showed a the nonsignificant difference in *atbzip62* and Col-0 WT at 0, 6, and 48 hpi (Figure 3B). Additionally, the electrolyte leakage results indicated that *atbzip62* mutant plants exhibited a lower electrolyte leakage over time compared with the examined genotypes except for Col-0 WT (Figure 3C). Therefore, owing to the observed transcript accumulation patterns of pathogenesis genes in *atbzip62* compared with Col-0 and disease-susceptible mutant lines, associated with the observed resistance phenotype over time of *atbzip62* against *Pst DC3000 vir*, *AtbZIP62* TF, earlier identified as a positive regulator of drought stress response in *Arabidopsis* and a negative regulator of SOS signaling pathway genes. So here, we proposed *AtbZIP62* TF as a negative regulator of basal defense against *Pst DC3000 vir* in *Arabidopsis.*

### 3.3. AtbZIP62 TF Negatively Regulates Systemic Acquired Resistance

Multicellular organisms, including plants, have developed a mechanism to systemically communicate the occurrence of a wound or an external stimulus to help them escape or defend themselves [28]. Signaling in plants during pathogens attack plays a crucial role in providing an alert to activate the necessary defense mechanism to combat the stress. However, one reaction of the plant to pathogen infection is said to be the induction of a long-lasting involvement of constitutive barriers known as SAR [22,39], characterized by an activation of a broad spectrum of host defense mechanisms, both locally at the site of earlier pathogen attack and systemically in tissues untouched by the pathogen [40]. Here, we observed that soon after *Pst DC3000 avrB* was inoculated, the transcript accumulation pattern of *AtPR1* and *AtPR2* significantly increased over time in distal leaves in *atbzip62* knockout plants compared with WT (Col-0) and in susceptible *atgsnor1-3* and *atsid2* plants (Figure 3A,B). Similarly, the SAR marker genes *AtAZI* and *AtG3DPH* were significantly upregulated in *atbzip62* under the same conditions. Previous studies reported that some members of the *bZIP* TF family are involved in the regulation of SA-mediated signaling of SA during the biotic stress response [41,42]. For instance, the *Arabidopsis tga7* mutant showed increased susceptibility to *Pseudomonas syringae* pv. *maculicola*, which suggested *AtTGA7* as being involved in the regulation of *PR* gene expression [43]. Therefore, our data suggested the *AtZIP62* TF as a negative regulator of basal defense and SAR in *Arabidopsis* in response to *Pst DC3000*.

## 4. Materials and Methods

### 4.1. Plant Materials and Growth Conditions

The seeds of the WT *Arabidopsis* Col-0 and those of the knockout line *atbzip62* (AT1G19490: SALK_053908C) were obtained from the *Arabidopsis* Biological Resources Center (https://abrc.osu.edu accessed on 20 March 2019). Another set of knockout lines, *atgsnor1-3*, lacking the *S-*nitrosoglutathione (GSNO) reductase 1 (*GSNOR1)*, which regulates cellular *S*-nitrosothiols (SNO) levels [27], and *atsid2*, known as an (SA-deficient mutant) [44], were included in the study as disease-susceptible control lines [27,45]. All genotypes used in the study were from the Col-0 genetic background. The *atbzip62* knockout was genotyped to identify homozygous transfer DNA (T-DNA) insertion lines, which was confirmed, as described earlier [12]. Before germination, seeds were surface-sterilized in 50% commercial bleach (1 mL) solution containing 0.1% Triton X-100 (Sigma Aldrich, USA) in a 1.5 mL Eppendorf tube (e-tube) for about 1–2 min by pipetting up and down, followed by rinsing with sterile distilled water up to five times, and incubated at 4 °C overnight, as described earlier [9]. Then, seeds were sown and grown on peat moss soil mixture at 22 °C, with 16 h light and 8 h dark cycles.

### 4.2. Pathogen Growth, Inoculation, and Pathogenicity Test

Before inoculating *Arabidopsis* plants, the biotrophic bacterial pathogen *Pst* strain *DC3000*
*vir* and *avrB* was grown (single colony) and maintained as described earlier [46]. The bacterial pathogen was grown on Luria–Bertani (LB) agar plates modified with rifampicin (50 μL/50 mL), followed by incubation at 28 °C in a shaking incubator. The overnight bacterial culture (1 mL) was centrifuged for three minutes at 8000 rpm to pellet down the cells, and the bacterial cells were resuspended in 1 mL of 10 mM of magnesium chloride (MgCl_2_), followed by reading the absorbance of the culture at a OD_600_ nm wavelength with MgCl_2_ used as blank [29].

Plants were inoculated by infiltrating the bacterial inoculum of 5 × 10^5^ CFU mL^−1^ (obtained after serial dilutions, OD_600_ = 0.002) in triplicate [9,47] using a 1 mL syringe (without the needle) into the abaxial side of the leaf (the lower leaf surface). Control plants (mock) were infiltrated with 10 mM of MgCl_2_. Leaf samples were collected at 0, 1, 2, and 3 dpi for gene expression analysis of *Arabidopsis* plants inoculated with *Pst DC3000 vir*. The phenotypic response of *Arabidopsis* genotypes was observed over time, up to 9 dpi, and photos were taken using a DSLR camera (EOS 700D, Canon).

For analyzing the pathogen growth, leaf discs collected from both *Pst DC3000 vir*-inoculated leaves using a cork borer (1 cm diameter) were homogenized (by crushing with a plastic pestle in a 1.5 mL e-tube) in 10 mM of MgCl_2_, and the supernatant was diluted 10 times. A hundred microliters (100 μL) was spread on LB agar plates containing antibiotics, followed by incubation at 28 °C for 72 h. The pathogen growth was determined by counting the bacterial CFU mL^−1^ as previously described [48].

### 4.3. Total RNA Extraction, cDNA Synthesis, and Real-Time qRT-PCR Analysis

Total RNA was isolated from leaf samples using the TRI-Solution^TM^ Reagent (Cat. No: TS200-001, Virginia Tech Biotechnology, Lot: 337871401001), as recommended by the manufacturer’s protocol. Thereafter, the complementary DNA (cDNA) was synthesized as described earlier [49]. Briefly, 1 µg of RNA was used to synthesize cDNA using BioFACT^TM^ RT-Kit (BioFACT^TM^, Daejeon, Korea) according to the manufacturer’s standard protocol. The cDNA was then used as a template in qRT PCR to study the transcript accumulation of selected genes (Appendix A) using a real-time PCR master mix including SYBR green (BioFact, Korea) along with 100 ng of template DNA and 10 nM of each forward and reverse primer in a final reaction volume of 20 µL. A no-template control was used as a control. A 2-step reaction, including polymerase activation at 95 °C for 15 min, followed by denaturation at 95 °C for 5 s, and annealing and extension at 65 °C for 30 s, was performed in a real-time PCR machine (Eco™ Illumina, San Diego, CA, USA). Total reaction cycles were 40 and the data were normalized with a relative expression of *Arabidopsis* Actin2.

### 4.4. In Silico Promoter Analysis and Prediction of Protein–Protein Interaction

Based on the transcript accumulation pattern of pathogenesis-related genes (*AtPR1* and *AtPR2*) and that of the SAR-marker genes (*AtAZI* and *AtG3PDH*) recorded in the *atbzip62* background, we were interested to investigate the mechanism underlying their transcriptional interplay with the *AtbZIP62* TF. To achieve that, we performed an *in silico* promoter analysis and a protein association network analysis using the STRING database (https://string-db.org/ (accessed on 9 October 2021)). In addition, the transcription regulation prediction was performed using the PlantRegMap feature within the Plant Transcription Regulatory Map (http://plantregmap.gao-lab.org/binding_site_prediction_result.php (accessed on 9 October 2021)). The DNA sequences (coding sequence, CDS, FASTA format: “>ATxGxxxxx” proceeds the sequence) of target genes used for the prediction of binding sites specific to bZIP TFs were downloaded from the *Arabidopsis* Information Resource database (TAIR, https://www.arabidopsis.org/index.jsp (accessed on 9 October 2021)).

### 4.5. Assessment of Systemic Acquired Resistance

The SAR was analyzed, using the method as previously described [10]. Plants were grown under the same conditions described in Section 4.1, and the *avrB* inoculum was prepared as shown in Section 4.2. Briefly, the bacterial inoculum was infiltrated with a syringe (1 mL) without the needle on the abaxial side of leaves (local leaves). However, leaf samples were obtained at 0, 3, 6, 12, 24, and 48 hpi from distal leaves in triplicate for gene expression analysis of SAR marker genes (*AtAZI* and *AtG3DPH*), as well as well-known pathogenesis-related genes, *AtPR1* and *AtPR2* [50].

### 4.6. Electrolyte Leakage Assay

To quantify the *Pst DC3000*-induced cell death or membrane injury, electrolyte leakage was conducted after pathogen infection as described earlier [9]. Briefly, 10 uniform local leaf discs (1 cm in diameter) harvested with a cork borer from different plants of each *Arabidopsis* genotype inoculated with *Pst DC3000 avrB* were taken, rinsed three times with deionized water, and floated in 5 mL of deionized water in a 6-well culture plate (SPL life sciences, Pocheon-si, Korea) for 30 min. The electrolyte leakage of each sample was recorded over time using a portable conductivity meter (HURIBA Twin Cond B-173, Kyoto, Japan).

### 4.7. Statistical Analysis

Experimental data were collected in triplicate and statistically analyzed using GraphPad Prism software (version 7.00, 1999–2016 GraphPad Software, Inc., San Diego, CA, USA). Analysis of variance was performed, and the least significant difference was calculated at the significance level of 0.05. To assess the statistical significance level of the observed changes in the expression of target genes between *Arabidopsis* genotypes, we compared the relative expression in the mutant lines with those observed in Col-0 WT.

## 5. Conclusions

The response mechanisms of plants toward bacterial pathogens infection involve the activation of different signaling components and metabolic pathways, tending to provide the plant with the needed level of resistance for its survival and fitness. In the process, positive and negative regulators of plant defense are induced or suppressed, and their interplay is of paramount importance to guarantee balanced cellular functioning. This study examined the role of *AtbZIP62*, a gene encoding the *bZIP* TF in the adaptive response mechanisms toward *Pst DC3000* infection. Results showed that when *Pst DC3000 vir* was inoculated (by infiltration), the transcript accumulation of *AtPR1* and *AtPR2* genes were increased significantly in *atbzip62* soon after inoculation compared to Col-0 and *atgsnor1-3* and *atsid2*, with *AtPR1* prevailing over *AtPR2*. However, upon inoculation with *Pst DC3000 avrB* in the local leaves of *Arabidopsis* genotypes, the transcript accumulation of *AtAZI* and *AtG3DPH*, two well-established SAR marker genes, was significantly upregulated in the systemic leaves of *atbzip62* mutant plants soon after inoculation compared to Col-0 WT, *atgsnor1-3*, and *atsid2.* Therefore, regarding the enhanced resistance phenotypic phenotype of the *atzbip62* line, coupled with the pathogen growth pattern and the increased transcript accumulation of analyzed PR related genes upon *Pst DC3000 vir* infection, associated with the upregulation of SAR marker genes in systemic leaves (under *avrB* inoculation). Hence, this study proposes the *AtbZIP62* TF as a negative regulator of basal defense and SAR in *Arabidopsis*.

## Figures and Tables

**Figure 1 ijms-22-11541-f001:**
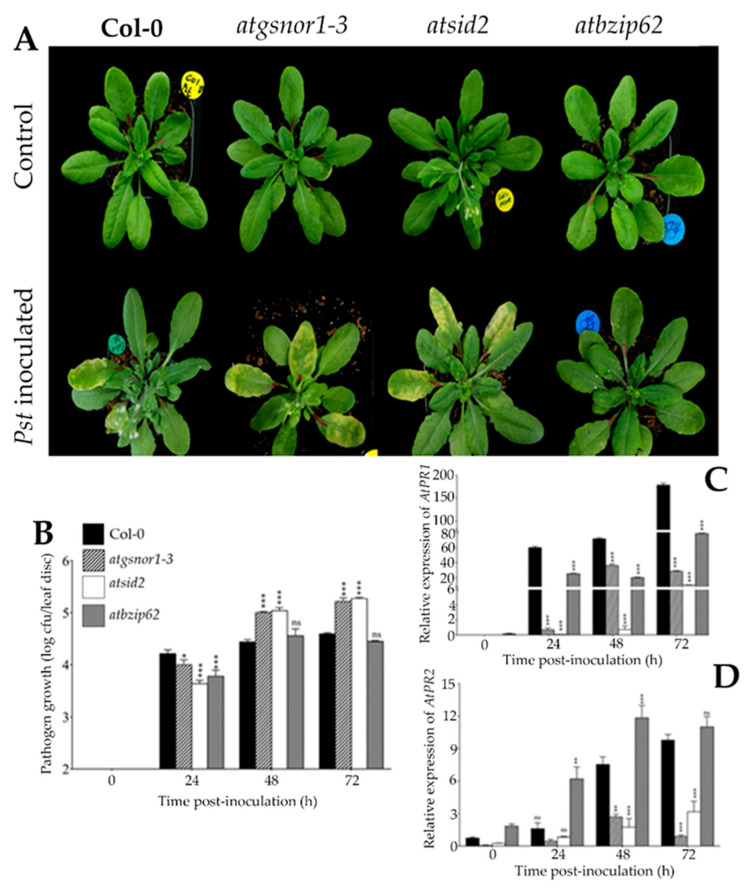
*AtbZIP62* negatively regulates basal defense. (**A**) Phenotypic response of *Arabidopsis* genotypes (Col-0 WT, *atgsnor1-3*, *atsid2*, and *atbzip62*) challenged with *Pseudomonas syringae* pv. *tomato* (*Pst*) *DC3000* infection; (**B**) growth pattern over time of *Pst DC3000 vir* in tested genotypes; (**C**) expression level over time of *AtPR1*; (**D**) expression level over time of *AtPR2* in response to *Pst DC3000 vir*. The phenotypes were recorded at 9 dpi, and leaf samples for gene expression analysis were collected over time (0, 24, 48, and 72 hpi) in triplicate. To confirm the phenotype, the experiments were repeated four times. * *p* < 0.05; ** *p* < 0.01; *** *p* < 0.001; ns, nonsignificant.

**Figure 2 ijms-22-11541-f002:**
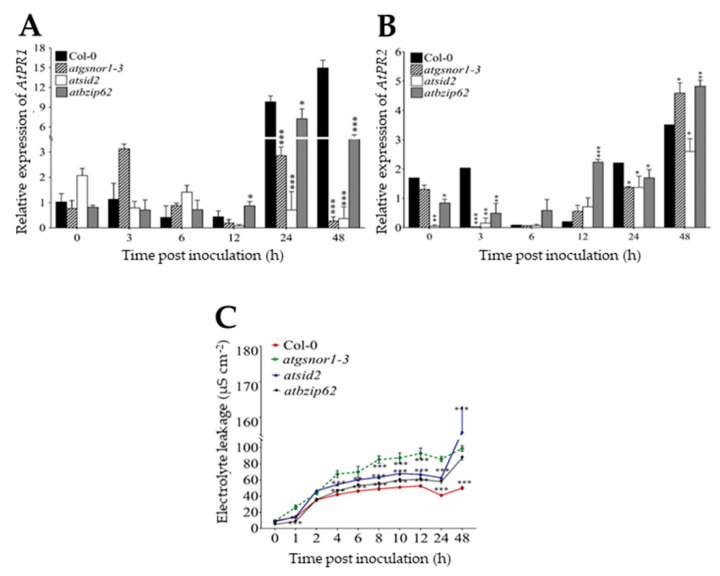
Transcript accumulation patterns of pathogenesis-related genes and electrolyte leakage results. Expression level over time of (**A**) *AtPR1* and (**B**) *AtPR2* in local leaf samples of Col-0 (WT), and *atgsnor1-3, atsid2*, and *atbzip62* mutant lines in response to *Pst DC3000 vir* for *R* genes-mediated resistance soon after bacterial inoculation. (**C**) Electrolyte leakage measured over time. Data are the mean expression values in triplicate normalized to the relative expression of *AtActin1*. * *p* < 0.05; ** *p* < 0.01; *** *p* < 0.001; empty are nonsignificant.

**Figure 3 ijms-22-11541-f003:**
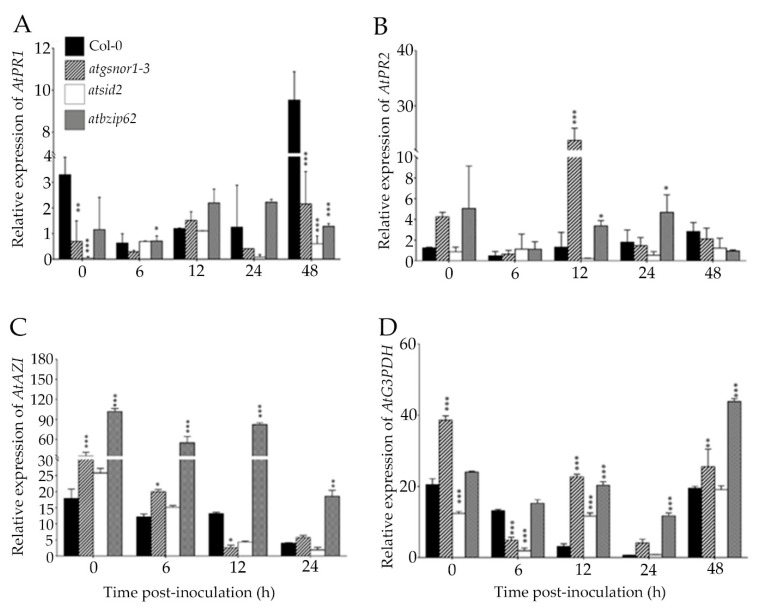
*AtbZIP62* negatively regulates SAR. Transcript accumulation of (**A**) *AtPR1*, (**B**) *AtPR2*, (**C**) *AtAZI*, and (**D**) *AtG3DPH* genes in the systemic leaves of Col-0 (WT), *atgsnor1-3*, *atsid2*, and *atbzip62* mutant lines under *Pst DC3000 avrB* inoculation. Data are the mean expression values in triplicate normalized to the relative expression of AtActin1. ** p* < 0.05; ** *p* < 0.01; *** *p* < 0.001; empty are non-significant.

**Figure 4 ijms-22-11541-f004:**
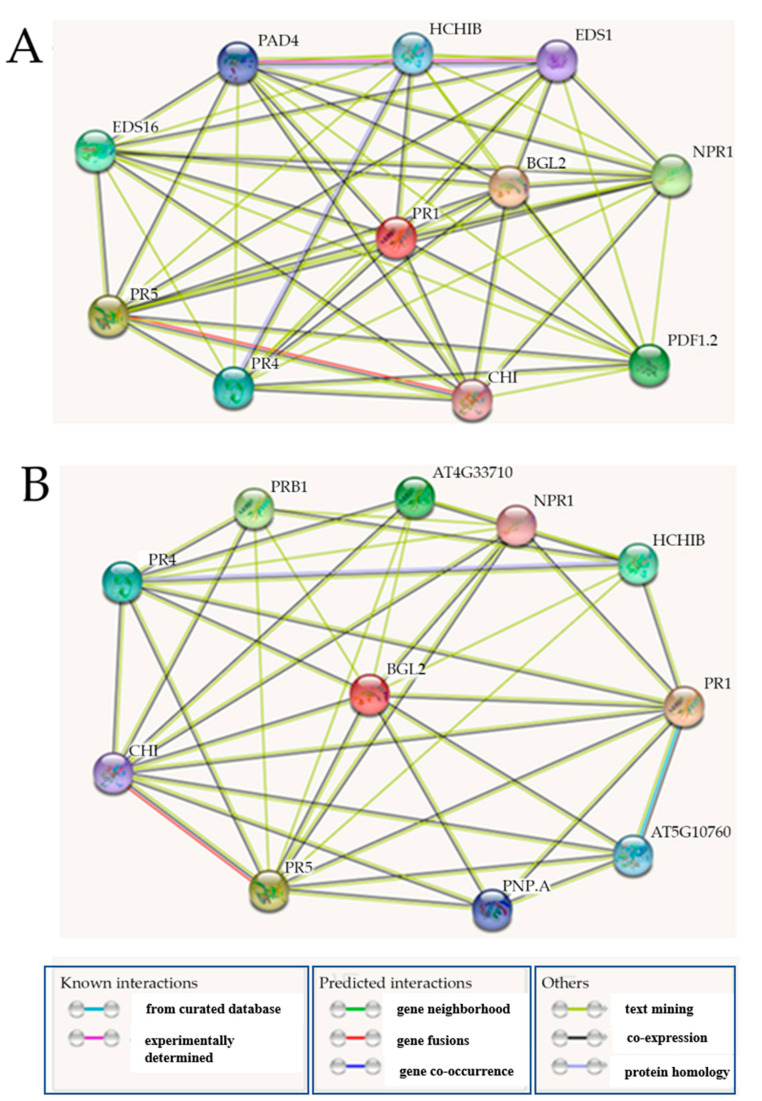
Prediction of the functional protein–protein interactome of selected pathogenesis-related genes. Predicted protein–protein interaction network involving the Arabidopsis (**A**) AtPR1 and (**B**) AtPR2 or BGL2 (proteins) with other pathogenesis-related protein-coding genes. The prediction was performed using the protein sequence of each target gene obtained from the Arabidopsis Information Resource (TAIR, www.arabidopsis.org (accessed on 9 October 2021)) and uploaded to STRING protein database version 11.5 (https://string-db-org/ (accessed on 9 October 2021)). Red nodes indicate the gene of interest, while other nodes represent the interactome of a different nature indicated by connecting lines. The nature of the interaction is indicated by specific line colors and the source of information (from curated databases, experimentally determined, predicted interactions, and others), as explained in the figure legend below the panel (**B**).

**Table 1 ijms-22-11541-t001:** Identified transcription factor binding sites in the promoter of target genes.

Gene Locus ID	TF Name	Target Genes	Position	Strand	*p*-Value	q-Value	Matched Sequence
** *AtPR1* **							
*	*	*AT2G14610*	*	*	*	*	*
*	*	*AT2G14610*	*	*	*	*	*
** *AtPR2* **							
AT1G06070	*AtbZIP69*	*AT3G57260*	81–91	+	2.8 × 10^−5^	0.0242	CACAGCTGGAC
AT1G06070	*AtbZIP69*	*AT3G57260*	80–90	−	2.9 × 10^−5^	0.0242	TCCAGCTGTGT
** *AtAZI1* **							
AT1G06070	*AtbZIP69*	*AT4G12470*	226–236	+	6.05 × 10^−6^	0.00491	AACAGCTGTCC
AT1G06070	*AtbZIP69*	*AT4G12470*	225–235	−	2.73 × 10^−5^	0.0111	GACAGCTGTTT
AT2G40620	*AtbZIP18*	*AT4G12470*	226–236	+	1.37 × 10^−5^	0.0112	AACAGCTGTCC
** *AtG3PDH* **							
*	*	*AT2G41540*	*	*	*	*	*
*	*	*AT2G41540*	*	*	*	*	*

(*) the asterisk indicates that the binding sites for the *AtbZIP18* and/or *AtbZIP69* associated with the *AtbZIP62* TF were not detected (http://plantregmap.gao-lab.org/binding_site_prediction.php (accessed on 9 October 2021)). The positive (+) and negative (−) signs indicate positive and negative orientation of DNA strands, respectively.

## Data Availability

Not applicable.

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
