# Peer review of "Enhanced Resistance of atbzip62 against Pseudomonas syringae pv. tomato Suggests Negative Regulation of Plant Basal Defense and Systemic Acquired Resistance by AtbZIP62 Transcription Factor"

_ijms, 2021, doi:10.3390/ijms222111541_

Round 1

Reviewer 1 Report

The author demonstrated the involvement of AtbZIP62 in negative regulation of basal defense and systemic acquired resistance in Arabidopsis. However, the paper only answers the question of what is this phenomenon. We still do not know why and how this protein works. Answering these questions will significantly improve the quality of your finding.

In my opinion, the possible interaction between this protein and host proteins may down-regulate the host defenses. Otherwise, this transcription factor possibly interacts with the promoter/regulator DNA regions or other transcription regulators to create a negative effect on the host defense genes expression. To this extent, further experiments including Protein-Protein and DNA-Protein interactions may be required. 

Author Response

We are thankful to the editorial office and anonymous reviewers for the time given to evaluate our manuscript. We appreciate their comments and are happy to share that most of the comments are addressed and have substantially improved the quality of the manuscript. We would like to specify that all changes in the manuscript were highlighted in green. We believe that the manuscript in the current form will be suitable for publication in the journal.

Author’s pointwise response to the review report (Reviewer 1)

Point 1: The author demonstrated the involvement of AtbZIP62 in negative regulation of basal defense and systemic acquired resistance in Arabidopsis. However, the paper only answers the question of what is this phenomenon. We still do not know why and how this protein works. Answering these questions will significantly improve the quality of your finding.

Response 1: We are thankful to the reviewer for all the comments and concerns raised to our attention to improve the quality of the manuscript. We would like to indicate that we have tried to include additional analysis and extend the discussion. Particularly on the possible DNA-protein or protein-protein interaction between the AtbZIP62 and the pathogenesis-related genes as well as those associated with SAR to better understand the mechanism underlying the observed AtbZIP62 TF-mediated suppression of plant defense in response to the bacterial pathogen Pseudomonas syringae pv. tomato.

Point 2: In my opinion, the possible interaction between this protein

Response 2: We appreciate the reviewer for his valuable comments. The authors agree that DNA-protein or protein-protein interaction assay would provide deep insights into the observed enhanced resistance phenotype of the atbzip62 mutant line, in addition to the expression pattern of pathogenesis-related genes in response to Pst DC3000 vir and avrB. We would like to indicate that we were also interested to investigate further this aspect of the functional characterization of AtbZIP62 encoding a transcription factor; however, our laboratory is currently having limited resources, majorly financial constraints, to perform protein-protein or DNA-protein interaction related experiments. We would like to apologize for this inconvenience, despite the high interest. Owing to the above, we conducted in silico promoter analysis and protein-protein interaction using a publicly available database. These databases offer useful information on potential cis-regulatory elements present in the promoter of the target genes (PlantPAN) and protein-protein interaction (string)-related assays. A new table of the promotor analysis and Figure 4 related to the prediction of protein-protein interaction between PR genes have been added to the manuscript and described accordingly.

Reviewer 2 Report

The introduction needs to be implemented

The results need to be write better, please separate the introduction from the data results, reprted in the results section.

Material and methods need to be improved

Author Response

Point 1: The introduction needs to be implemented

Response 1: We are thankful to the reviewer for his valuable comments and suggestions to improve the quality of the manuscript. We have revised the manuscript thoroughly as suggested. We are happy to share that almost all the comments have been addressed in the revised version of the manuscript

Point 2: The results need to be write better, please separate the introduction from the data results, reported in the results section.

Response 2: The results section has been improved as per the suggestions by the reviewer.

Point 3: Material and methods need to be improved

Response 3: The section materials and methods have been improved to give more clarity on the methodology used in the study.

Reviewer 3 Report

The manuscript by Nabi et al. on “Enhanced Resistance of atbzip62 against Pseudomonas syringae 2 pv. tomato Suggests Negative Regulation of Plant Basal Defense and Systemic Acquired Resistance by AtbZIP62 Transcription Factor” studied the involvement of atbzip62 in conferring resistance to Pst in Arabidopsis. I hope the following comments will help in improving the quality of the manuscript.

Major comments:

  1. In section 1. Enhanced Phenotypic Resistance of atbzip62 toward Pst DC3000 vir, both Col-0 and atbzip62 showed comparable CFUs of Pseudomonas syringae and relative expression of AtPR1 and AtPR2. With the given pathogen concentration and relative PR-genes expression how can it be concluded that atbzip62 plays role in Col-0 line and in conferring Pst resistance? This context also is present in the following results.
  2. It is interesting to see the levels of expression of AtPR1 in Col-0 at 0 hpi for the experiments in figure 2 and 3. Why would the gene be so highly expressed at 0 hpi in figure 3A and not so much in figure 2A? In figure 3A, although the levels of the gene expression in Col-0 is not at its peak at 12 and 24 hpi in figure 3A, it is significantly high at 48 hpi. Does this play a role in SAR and how can it be related with atbzip62?
  3. qPCR procedure is completely missing from the methods section. How were relative gene expressions performed?
  4. There are numerous writing errors in the manuscript which needs to be paid close attention and corrected. Some of the errors have been pointed out below however, by no means the highlighted errors should be considered the complete list.

Minor comments:

Line 59: Use of different terminology for “antioxidant systems” is suggested. Are not ROS responses involved in basal (PTI) and R-gene (ETI) as well?

Line 61: Confirm if PTI also confers hypersensitive response and rephrase the sentence accordingly.

Line 66: Use another word for “susceptible” in “It is said that PTI is susceptible to suppression …”

Line 70. “However, …” this sentence does not connect very well to the previous sentences. Please rephrase.

Line 106: inoculated in atbizip62 plants

Line 146: Even basal defense leads to downstream defense signaling and long-lasting resistance initiation

Figure 2C: Confirm that there are no two data points for the same hpi for atsid2

Line 283: 16 h light

Line 301: cork borer? Also in line 317

Line 304: Confirm the incubation time: 72 days or hours?

Line 310: Explain what is being referred by local leaves? Also in line 317.

Author Response

The manuscript by Nabi et al. on “Enhanced Resistance of atbzip62 against Pseudomonas syringae 2 pv. tomato Suggests Negative Regulation of Plant Basal Defense and Systemic Acquired Resistance by AtbZIP62 Transcription Factor” studied the involvement of atbzip62 in conferring resistance to Pst in Arabidopsis. I hope the following comments will help in improving the quality of the manuscript.

Response: We are thankful to the reviewer for all the comments and concerns raised to our attention to improve the quality of the manuscript.

Major comments:

Point 1: In section 1. Enhanced Phenotypic Resistance of atbzip62 toward Pst DC3000 vir, both Col-0 and atbzip62 showed comparable CFUs of Pseudomonas syringae and relative expression of AtPR1 and AtPR2. With the given pathogen, concentration and relative PR-genes expression how can it be concluded that atbzip62 plays role in Col-0 line and in conferring Pst resistance? This context also is present in the following results

Response 1: We would like to thank the reviewer for this valuable concern raised. We initially expected to see a significant reduction in pathogen growth in the atbzip62 mutant line after observing its resistance phenotype compared to that of Col-0 and the susceptible reference genotypes against Pst DC3000 vir. However, in this study, the growth of the pathogen was shown to decrease soon after pathogen inoculation but later on, Col-0 and the mutant showed similar pathogen growth patterns.

We could say that…

The AtbZIP62 plays its role at the initial perception and onset of the pathogen. In the absence of AtbZIP62, the pathogen growth was significantly limited within the first 24hrs of inoculation of the pathogen (Figure 1B) while later on both Col-0 and atbzip62 had comparable growth of the pathogen and that is why a stationary phase of the pathogen was observed at 48h and 72h post-inoculation. However, the highly susceptible mutants i.e. atgsnor1-3 and atsid2 were still in the log-phase of the pathogen growth (Figure 1B). The data reveals that the log-phase of the pathogen growth is also important right from the earlier perception of the pathogen for symptoms development as evident in Figure 1C, where atbzip62 reveals no symptoms while Col-0 had mild, atgsnor1-3, and atsid2 had severe symptoms.

On the other hand, the PR gene expression also seconds the earlier data that the AtPR2 expression level is significantly higher at the earlier time points (24 h and 48 h) in the atbzip62 as compared to Col-0 and the other susceptible mutants (Figure 1D).  Later on, at 72 h post-inoculation of the pathogen, where PstDC3000 was at stationary growth-phase in Col-0 and atbzip62 (Figure 1B), the AtPR2 expression was also non-significant among the two (Figure 1D).  These findings can be concluded as, that AtbZIP62 negatively regulates the basal defense at earlier time points by favoring the growth of the pathogen and limiting the AtPR2 gene expression.

Point 2: It is interesting to see the levels of expression of AtPR1 in Col-0 at 0 hpi for the experiments in figure 2 and 3. Why would the gene be so highly expressed at 0 hpi in figure 3A and not so much in figure 2A? In figure 3A, although the levels of the gene expression in Col-0 is not at its peak at 12 and 24 hpi in figure 3A, it is significantly high at 48 hpi. Does this play a role in SAR and how can it be related with atbzip62?

Response 2: We appreciate the concern raised by the reviewer. We would like to indicate that Figure 2A refers to R-gene mediated defense. Whereas, Figure 3A shows systemic acquired resistance (SAR), the expression of the pathogenesis-related genes was measured in inoculated leaves (local leaves) and systemic or distal leaves (unwounded leaves of the same plant positioned away from the local leaf to investigate the activation of defense-related genes and the SA-signaling pathway as well as the SAR.

Point 3: qPCR procedure is completely missing from the methods section. How were relative gene expressions performed?

Response 3: We would like to apologize for the inconvenience. We have included a new subsection 4.3 describing the qPCR analysis.

Point 4: There are numerous writing errors in the manuscript which needs to be paid close attention and corrected. Some of the errors have been pointed out below however, by no means the highlighted errors should be considered the complete list.

Response 4: We would like to apologize for the writing errors in the manuscript. We have revised the manuscript for grammatical errors and typos as suggested by the reviewer.

Minor comments:

Point 1: Line 59: Use of different terminology for “antioxidant systems” is suggested. Are not ROS responses involved in basal (PTI) and R-gene (ETI) as well?

Response 1: We are thankful to the reviewer the concern. We have improved the introduction, taking into account the concern raised.

Point 2: Line 61: Confirm if PTI also confers hypersensitive response and rephrase the sentence accordingly.

Response 2: ​We have modified the introduction section, and we have provided additional information while taking into account the concern raised by the reviewer.​

Point 3: Line 66: Use another word for “susceptible” in “It is said that PTI is susceptible to suppression …”

Response 3: We have replaced the word “susceptible” with another more convenient word as suggested by the reviewer.

Point 4: Line 70. “However, …” this sentence does not connect very well to the previous sentences. Please rephrase.

Response 4: We have modified this paragraph, and the concern raised by the reviewer has been addressed.

Point 5: Line 106: inoculated in atbizip62 plants

Response 5: The introduction section has been reviewer thoroughly. Many of the concerns raised have been addressed.

Point 6: Line 146: Even basal defense leads to downstream defense signaling and long-lasting resistance initiation

Response 6: We agree with the reviewer. We have modified this statement accordingly.

Point 7: Figure 2C: Confirm that there are no two data points for the same hpi for atsid2

Response 7: We would like to thank the reviewer for the concern. We hereby confirm that there is no two data points for the same hpi for atsid2 in Figure 2C. The gap we see cutting the line is created by the break of the y-axis. The axis break is done to allow the display of data with big value gaps among groups to be displayed in the same plot.

Point 8: Line 283: 16 h light

Response 8: We appreciate the concern raised by the reviewer. we have corrected accordingly.

Point 9: Line 301: cork borer? Also in line 317

Response 9: We would like to thank the reviewer for the concern. We have corrected the typos as recommended.

Point 10: Line 304: Confirm the incubation time: 72 days or hours?

Response 10: We apologize for the inconvenience. We intended to write 72 h, not days. We have corrected it accordingly.

Point 11: Line 310: Explain what is being referred by local leaves? Also in line 317.

Response 11: ​The use of local leaves has been described in the manuscript as inoculated leaves. While distal leaves are also referred to as systemic leaves. We have harmonized the terminology in the manuscript to avoid any form of misinterpretation.

Round 2

Reviewer 2 Report

the manuscript, in this new version, is improved

Author Response

We would like to thank you for the reviewer's kind suggestions towards the improvement of our MS. We highly appreciate the worthy reviewer comments and suggestions. 

Reviewer 3 Report

I would like to thank the authors for making the necessary changes in the manuscript and providing convincing response to the comments. I do not have any further major concerns. 

Please provide high resolution picture for figure 4.

Please pay attention to grammatical errors present in the manuscript.

Author Response

We would like to thank you for the reviewer's kind suggestions towards the improvement of our MS. We highly appreciate the worthy reviewer comments and suggestions. 

We have checked the grammatical errors and improved the Figure 4 resolution as per the reviewer's suggestion. Thank you. We hope that current form of MS is suitable for publications.